# The Role of Well-Being, Divergent Thinking, and Cognitive Reserve in Different Socio-Cultural Contexts

**DOI:** 10.3390/brainsci15030249

**Published:** 2025-02-26

**Authors:** Francesca Garau, Alessandro Antonietti, Natale Salvatore Bonfiglio, Beatrice Madeddu, Maura Crepaldi, Jessica Giannì, Giulia Fusi, Laura Colautti, Virginia Maria Borsa, Massimiliano Palmiero, Maria Luisa Rusconi, Maria Pietronilla Penna

**Affiliations:** 1Department of Pedagogy, Psychology, Philosophy, University of Cagliari, 09127 Cagliari, Italy; f.garau91@gmail.com (F.G.); beamadeddu@gmail.com (B.M.); penna@unica.it (M.P.P.); 2Department of Psychology, Università Cattolica del Sacro Cuore, 20123 Milan, Italy; laura.colautti1@unicatt.it; 3IRCCS Centro San Giovanni di Dio Fatebenefratelli, 25125 Brescia, Italy; nbonfiglio@fatebenefratelli.eu; 4Department of Human and Social Sciences, University of Bergamo, 24129 Bergamo, Italy; maura.crepaldi@unibg.it (M.C.); jessica.gianni@unibg.it (J.G.); giulia.fusi@unibg.it (G.F.); marialuisa.rusconi@unibg.it (M.L.R.); 5Department of Communication Sciences, University of Teramo, 64100 Teramo, Italy; mpalmiero@unite.it

**Keywords:** well-being, creativity, cognitive reserve, divergent thinking, elders, Blue Zones

## Abstract

Background: Many protective factors promote psychological well-being (PWB) in the elderly and contribute to healthy aging, such as engagement, positive emotions, and cognitive reserve (CR), which includes education, leisure, and work activities. CR sustains cognitive functioning and positively correlates with creativity, particularly divergent thinking (DT), helping older adults cope with everyday challenges and enhancing their PWB. Objectives: The present study aimed to investigate the relationships between DT, CR, PWB, memory functions, depression, stress, and anxiety indexes even in the Blue Zone (BZ), an area known for extraordinary longevity and high PWB. Methods: A total of 165 Italian healthy older adults (Mage = 73.8, SD = 6.7) from Milan (MI), Bergamo (BG), Cagliari (CA), and BZ were enrolled and divided into four groups according to their origin. Generalized linear models (GLMs) with normal and gamma link functions were used. Results: BZ presented higher DT and PWB indices than the cities but lower CR, particularly in education. Conclusions: This study highlights the influence of DT in supporting cognitive functions and PWB, suggesting that PWB and DT are key protective factors in aging.

## 1. Introduction

Blue Zones (BZs) are small, mountainous regions isolated from other population centers, distinguished by extraordinary longevity—both in terms of the proportion of long-lived inhabitants and the psycho-physical well-being of the population [1,2]. These areas include Ikaria (Greece), Okinawa (Japan), the central–eastern region of Sardinia (Ogliastra and Barbagia, Italy), and the Nicoya Peninsula (Costa Rica) [1]. Numerous researchers have studied the predictive factors of their extraordinarily successful aging. Findings, to date, suggest that environmental and contextual factors play a predominant role (75–80%), including geography, diet, lifestyle, and social connectedness [3], while genetic inheritance accounts for a smaller influence (20–25%) [4]. BZs are characterized by specific environmental features such as altitude, slope, and low pollution levels, alongside a simple and traditional way of life. This lifestyle is predominantly agro-pastoral, closely connected to nature, and involves significant physical activity [1,3,4,5,6,7]. From a psychological and social perspective, these areas exhibit collectivist societies [8], marked by stable and emotionally significant interpersonal relationships involving older adults in the social fabric. This fosters higher psychological well-being (PWB) and lower levels of depression [8,9,10,11]. Among the factors contributing most to the sense of well-being, the neighbourhood plays a key role as an enduring source of social and emotional support, complementing the support provided by one’s immediate family [1,12,13,14,15,16]. Furthermore, higher levels of perceived well-being and lower depressive indices are often accompanied by better cognitive functioning [8].

Despite these insights, the exact recipe for successful aging remains elusive. The primary factors identified so far include a healthy and engaged lifestyle, cognitive reserve (CR), and physical activity. BZs remain a key resource for understanding the impact of physical activity, engaged lifestyles, and psychological and social factors on successful aging [17]. Regarding lifestyle engagement, several studies have highlighted the importance of CR in fostering PWB and successful aging. CR supports rehabilitating neuronal pathways damaged by brain injury [18,19,20], playing the role of a protective factor during aging. However, most studies have focused on patients with brain injuries, limiting broader conclusions. More recent salutogenic studies have examined the relationship between quality of life and CR [21], finding that CR is associated with higher quality of life and moderates cognitive functioning and depressive symptoms [22,23]. For instance, Fastame, Brandas, and Pau [24] investigated the mediating role of CR in motor efficiency (through objectively assessing physical reserve) and PWB in the Sardinian BZs. The authors used the Geriatric Depression Scale (GDS) [25,26] to assess depressive symptoms and the Cognitive Reserve Index questionnaire (CRIq) [27] to assess CR as a factor shaped by lifestyle engagement, education, occupation, and leisure activities. Their findings showed that older adults with lower CR had higher depressive symptoms and poorer motor skills than those with higher CR. A year later, Fastame, Brandas, and Pau [28] explored physical reserve as a mediator between CR and global cognitive functioning, which was assessed using Folstein’s Mini-Mental State Examination (MMSE) [29] and incorporating perceived physical health as a variable. Their results demonstrated significant associations and supported the protective role of CR and physical reserve, which accounted for 18% and 32% of the variance in global cognitive functioning, respectively [30,31,32]. These studies affirmed the value of maintaining an active lifestyle, engaging in moderate but regular physical activity, participating in cognitively stimulating leisure activities to promote successful aging [33,34,35].

Research on factors promoting well-being has demonstrated the significant influence of CR on PWB, with higher CR linked to a higher quality of life in older adults. Stern’s research underscores that individuals with higher CR can optimize outcomes by utilizing alternative brain networks, which facilitate the development of adaptive strategies [18,19,20]. Recent studies have also identified a link between CR and creative thinking, particularly divergent thinking (DT), which is considered a core process of creative thinking, both of which may be preserved during aging [36] and contribute to cognitive functioning, with beneficial effects in both healthy and clinical populations [36,37,38,39,40,41]. Creative thinking refers to the ability to generate unexpected and original ideas or products that are useful and appropriate to the context [42]. It also involves the capacity to override automatic responses to develop alternative strategies, particularly in novel situations, and to navigate the challenges of everyday life [39,40,43,44]. Guilford [45,46] defined DT as the ability to think freely and explore multiple potential solutions to a problem. Recent studies have further emphasised a connection between CR and DT [47,48,49,50], suggesting that creative thinking, by fostering the development of adaptive skills, can enhance both psychological resources (such as self-efficacy and coping strategies) and cognitive processes [51,52]. As DT allows older adults to adopt alternative strategies when conventional approaches prove ineffective in tackling daily challenges [53], some researchers have proposed DT as an indicator or proxy for CR in aging populations [50,53,54,55]. The ability to think divergently—that is, to generate multiple solutions to an open-ended problem [56]—is a complex construct that integrates both crystallized intelligence components (such as semantic and autobiographical memory, which provides a foundation for new ideas) and fluid processing components (such as executive functions), which support semantic associations, inhibit automatic thinking, and allow for flexible attentional shifts [57]. As a result, DT (in particular, verbal components of DT predict CR [53]) serves primarily as a cognitive measure that helps assess an individual’s creative potential [23,24] and its role in CR, with significant implications for maintaining psychological and cognitive well-being throughout the aging process. Fusi and colleagues argued that DT should be recognized as a protective factor in aging, as it complements CR in positively influencing PWB [58]. This perspective aligns with Baltes’ SOC (Selection, Optimization, Compensation) model [59], highlighting the ability to employ internal strategies to achieve goals beneficial to psychological and emotional well-being [47,60]. Consequently, DT and CR share overlapping characteristics [53,54], making DT, as well as CR [61], one of the cognitive skills that could be preserved and increased to achieve PWB in aging [58].

However, it has not yet been determined what the trend of DT capacity is in the elderly population. In fact, the construct is multidimensional and difficult to define through a single comprehensive index [62]. While Colautti and colleagues observed that DT may be affected by age [63], other studies indicated that DT may remain intact, particularly when tasks are free from time constraints, cognitive workloads are manageable, and task materials engage diverse cognitive processes (e.g., verbal or visuospatial) [62,64]. Thus, DT emerges as a latent ability in aging, complementing CR as a contributor to successful aging [36]. Despite this, no study to date has specifically explored the role of creativity in successful aging within BZs—a gap our research aimed to address. Although no evidence yet links directly DT to PWB, some studies highlighted DT has an indirect positive effect on PWB through CR [53,58].

### 1.1. What Is Missing?

Due to the growing proportion of elderly people within the global population and the importance of developing interventions that help older adults reach the end of life free from physical and cognitive disabilities [65,66,67], this study aimed to investigate whether protective factors (such as DT and CR), which have been underexplored in BZs, could affect aging outcomes in these regions as they do in other contexts. BZs are known for their exceptional longevity and successful aging, making them ideal settings for such investigations. Previous research highlighted significant contextual differences between BZs and other areas [16,68]. For example, inhabitants of Northern Italian and non-rural areas exhibit higher depressive indices and lower PWB compared to those living in BZs [68]. This underscores the need to explore territorial differences in CR and DT. Notably, to the best of our knowledge, creativity—measured through indices such as fluency, flexibility, originality, and elaboration—has never been assessed in BZs. Our study represents the first attempt to evaluate creativity in this context, using validated tools to ensure replicability.

Evidence already suggests that DT predicts CR and mediates a positive indirect effect on PWB through CR [58]. Furthermore, previous studies have documented relationships between CR and PWB [24,54] and CR and global cognitive functioning [24], as well as DT, cognitive functioning, and CR [57,63,64,69,70,71]. CR has traditionally been assessed using socio-behavioral indicators, such as years of formal education and time spent on socio-cultural activities during the study period [33,35]. Research has recently begun to evaluate CR in BZs using validated tools encompassing work history, educational attainment, and leisure activities [24,28].

In terms of PWB, we investigated variables such as emotional competence, coping strategies, and personal satisfaction since previous studies have identified relationships between CR, DT, emotional competence [54,58], and coping strategies [58]. For depressive indices, we examined dimensions of stress, anxiety, and depression itself, recognizing the negative relationship between depressive indices and CR observed in other research [24]. We focused on fluid components of intelligence such as short-term memory, processing speed, visual scanning, number sequencing skills, shifting, and cognitive flexibility. It is important to note that the relationships between CR, cognitive functioning, and DT have been previously explored [62,63,70], indicating DT’s role in mitigating cognitive deficits [70]. However, its beneficial effects appear limited to crystalized cognitive components rather than fluid ones [63].

### 1.2. Questions

Our study investigated: (1) whether there are differences in a BZ as compared to different contexts in the same country—such as Milan (MI), Bergamo (BG), and Cagliari (CA)—in the components of CR and DT and in the variables already investigated, such as PWB, depressive indexes, and cognitive functioning; (2) whether these variables are related to each other within each different socio-cultural group (MI, BG, CA, and BZ).

## 2. Materials and Methods

### 2.1. Participants

A total of 179 participants were initially recruited for the study. Individuals with a missing Mini-Mental State Examination (MMSE) [72] score or a corrected MMSE score below 24 were excluded from the analyses. After applying these criteria, the final sample consisted of 165 participants (females = 92) with a mean age of 73.8 years (SD = 6.7). This exclusion ensured that the analyses concerned individuals with adequate cognitive functioning, as assessed by the MMSE. Table 1 shows the socio-demographic characteristics of the participants, divided by group.

Within the recruited sample, there were no significant differences in the number of participants across groups (*p* = 0.273), with the sample sizes ranging from 32 in CA to 50 in MI. Moreover, the mean age of participants was similar across groups and did not differ significantly (*p* = 0.72). The mean age ranged from 72.75 years (SD = 6.66, range = 25 years) in CA to 74.41 years (SD = 6.80) in BZ.

Also, gender distribution showed no significant differences among the groups (*p* = 0.061). The proportion of male participants ranged from 7.3% in the BZ to 12.7% in MI and BG. Female participants accounted for the majority in each group, with the highest representation in the BZ (17.6%).

However, significant differences were observed in the average years of education across groups (*p* = 0.024). Participants from CA (mean = 12.1 years, SD = 6.9) reported the highest educational attainment, followed by MI (mean = 11.3 years, SD = 4.81) and BG (mean = 10.33 years, SD = 3.2). Participants from BZ had the lowest average years of education (mean = 8.61 years, SD = 2.9).

### 2.2. Procedures

The sample was recruited according to the group (MI, BG, CA, BZ). Tests were administered individually during an in-person session, lasting approximately 90 min, to evaluate cognitive abilities, CR, DT, PWB, and emotional condition. All participants took part in the study voluntarily and without receiving any form of incentive.

The study was conducted in compliance with the ethical standards of the institutional research committee and the 1964 Helsinki Declaration [73]. It received approval from the Institutional Ethical Committee of the University of BG and the Ethical Committee of the Università Cattolica del Sacro Cuore in MI. Written informed consent was obtained from all participants to confirm their willingness to take part in the study.

### 2.3. Instruments

#### 2.3.1. Global Cognitive Functioning

MMSE [72] is a rapid-administration instrument used to assess global cognitive functioning. It consists of 30 items that examine seven cognitive domains: spatio-temporal orientation, motor coordination, attention, language, mental calculation, and short- and long-term memory. The maximum score is 30, with a cut-off of 24, below which a deficient performance is reported. The MMSE demonstrated adequate internal consistency (Cronbach’s alpha above 0.71), high test–retest reliability (ranging from 0.80 to 0.89), and good inter-rater reliability (0.75).

#### 2.3.2. Neuropsychological Testing

The digit span, in forward (Digit_FW) and backward (Digit_BW) versions [74], is a measure of verbal short-term memory. Specifically, Digit_FW assesses passive memory components without data elaboration, whereas Digit_BW evaluates active components, involving data elaboration. Participants listen to a sequence of numbers and have to repeat it immediately afterwards in the same order (Digit_FW) or in the reverse order (Digit_BW). The number of digits in the sequence gradually increases until it reaches a maximum of nine digits for Digit_FW and eight digits for Digit_BW. One point is awarded for each correct sequence and the test stops when the participant fails to repeat two sequences containing the same number of digits. The digit span tools showed internal consistency values ranging from 0.50 to 0.90.

Trail Making Test (TMT) [75] is composed of two parts (part A and part B), which allow for investigation of the spatial planning capacity and processing speed in a visual-motor task. Part A (TMT-A) estimates the speed of processing, visual scanning, and number sequencing skills: This task requires connecting 25 encircled numbers scattered on a white sheet with a continuous line in consecutive order, as fast as possible. Part B (TMT-B) is a measure of shifting and cognitive flexibility: Respondents have to draw a line between numbers and letters in an alternating sequential mode (e.g., 1-A-2-B-3-C), again as fast as possible. The final score is calculated by subtracting the time taken (measured in seconds) to complete part A from the time required to complete part B (i.e., TMT-B-A). The test–retest reliability was high for each score.

#### 2.3.3. Cognitive Reserve Index Questionnaire

CRIq [27] is a 20-item questionnaire that assesses the amount of CR acquired during a person’s lifetime. It is divided into three sections corresponding to the components of CR: educational attainment (CRIq School; CR_S), work experience (CRIq Work; CR_WR), and leisure activities (CRIq Leisure; CRIq_LA). The questionnaire investigates the type and frequency (weekly, monthly, yearly, and fixed) of activities conducted. Three different scores are used to quantify the CR related to school, work, and leisure activities. This tool, through the sum of the three indices, also provides a total measure of CR (CRIq-tot). Following the procedure suggested by the authors, the scores obtained are classified as low (≤70), medium–low (70–84), medium (85–114), medium–high (115–129), and high (≥130). The CRIq has a Cronbach’s alpha ranging from 0.62 to 0.73 in Italian population.

#### 2.3.4. Parallel Line Test (Torrance Tests of Creativity Thinking)

Parallel line test (LT), extrapolated from the Torrance Tests of Creativity Thinking (TTCT) battery [76], is used to evaluate figurative DT. The respondent is asked to draw multiple objects/things using straight-line pairs (30 pairs) within 10 min. The score is obtained, according to the manual, by considering three indices: fluidity (number of reliable answers provided), flexibility (number of semantic categories to which the answers belong), and originality (rarity of the answers: zero point = responses provided by ≥5% of 500 people; one point = responses provided by 2–4.99% of 500 people; two points = responses provided by <2% of 500 people or responses not in the manual). The Torrance Tests of Creative Thinking–Figural (TTCT-F) showed acceptable composite reliability (Ω = 0.81), as indicated by McDonald’s Omega coefficients.

#### 2.3.5. Psychological Well-Being

BEN-SSC [77] is a questionnaire for assessing PWB in adults and elderly people. The questionnaire consists of 37 items divided into three subscales based on the eudemonic concept of PWB proposed by Carol Ryff [78,79]: personal satisfaction (PWB_PS), coping strategies (PWB_CS), and ability to regulate emotions or emotional competence (PWB_EC). Respondents are asked to answer items using a four-point Likert scale (which goes from 1 = never to 4 = often/always). All items are self-descriptive and formulated to assess positive rather than negative attitudes and beliefs. The score is calculated for each subscale (PWB_PS, PWB_CS, PWB_EC) and, by adding up the answers of all items, the total score is obtained. The scores are divided into three levels: low (≤103), medium (104–114), and high (≥115). The higher the score, the higher the level of PWB. The reliability of the questionnaire was high (Cronbach’s alpha = 0.910).

Depression anxiety stress scale (DASS-21) [80] is a scale composed of 21 items investigating symptoms concerning three subdimensions: depression, anxiety, and stress. The person is asked to indicate, through a four-point Likert rating scale (ranging from 0 = “It has never happened to me” to 3 = “It happened to me most of the time”), the frequency at which each symptom occurred in the last week. A total score representing the general level of distress is provided by summarizing all responses. The DASS has a Cronbach’s alpha ranging from 0.74 to 0.90 in community samples from the Italian population.

### 2.4. Statistical Analyses

Although the sample size was relatively small, methodological precautions were taken to ensure the robustness of the results. Generalized linear models (GLMs) were used to explore the relationships between the dependent variables and the grouping factor, including robust covariance estimations to account for potential heteroscedasticity. These adjustments helped mitigate the impact of sample size limitations and improved the reliability of the statistical inferences. Each dependent variable was modeled separately to account for its specific characteristics. The assumption of normality was tested for all dependent variables by visually inspecting the residuals using Q–Q plots.

For variables exhibiting normal distribution, additional assumptions were assessed to ensure the validity of the GLMs (e.g., homoscedasticity and collinearity/linearity) through graphical diagnostics and statistical tests to ensure the robustness and interpretability of the models.

GLMs with an identity link function and a normal distribution were applied for variables exhibiting normal distributions. For variables where normality was not upheld, the alternative gamma distribution was used with either an identity or log-link function based on the nature of the variable and the results from the distribution fitting process. Coefficients were reported based on raw values, except for DASS scales where coefficients were reported in log-odds. The grouping factor (groups) was included as the main predictor. When CRIq subscales were used as predictors, models were also controlled for age. Maximum likelihood estimation (MLE) was used to estimate model parameters, with robust covariance matrices applied to manage potential heteroscedasticity in the data. Pairwise comparisons of estimated marginal means were conducted to explore group differences, with Bonferroni adjustments. Mean age was fixed at 73.7 in CR comparisons. Confidence intervals were calculated at a 95% level and Wald statistics were used to assess significance.

Spearman’s correlations were also used to evaluate the association between variables.

Chi-square test was used to compare frequencies across groups to evaluate sociodemographic differences among the four groups. For continuous variables, including age and mean years of education, Kruskal–Wallis test was applied to assess differences.

Significance level was fixed at 0.05. Confidence intervals were calculated at 95%. The SPSS (version 29.0.1) was used for GLM models. The R package (version 4.4.1; https://www.r-project.org/) was used to verify fitting distributions [81].

## 3. Results

### 3.1. Spearman’s Correlation Coefficients

To investigate the relationships between DT, CR, PWB, and emotional condition across groups, the associations between these variables were analyzed separately for BZ, MI, CA, and BG. This analysis aimed to highlight the unique patterns in BZ while identifying overlaps with BG, particularly in mechanisms associated with successful aging. Below, all significant correlations for each group are detailed (see Appendix A). It is important to note that the correlations identified in this study do not establish causal relationships between the examined variables. The observed associations suggest potential links, but further research is needed to determine the direction and nature of these relationships. Future studies employing longitudinal designs or mediation/moderation analyses could provide deeper insights into the underlying mechanisms and better clarify how these variables influence each other over time.

#### 3.1.1. Blue Zone

As reported in Table A1 (see Appendix A), significant positive correlations have been found between CR_WR with PWB_EC (ρ = 0.46, *p* ≤ 0.01), PWB_CS (ρ = 0.33, *p* ≤ 0.05), and between CR_LA and PWB_EC (ρ = 0.37, *p* ≤ 0.05). Moreover, significant negative correlations were found between CR_WR with both TMT_A (ρ = −0.31, *p* ≤ 0.05) and TMT_B (ρ = −0.32, *p* ≤ 0.05), but positive correlations were found between Digit_FW with CR_WR (ρ = 0.48, *p* ≤ 0.01) and CR_S (ρ = 0.31, *p* ≤ 0.05). Regarding DT, significant positive correlations were found between fluency with CR_S (ρ = 0.31, *p* ≤ 0.05), CR_WR (ρ = 0.44, *p* ≤ 0.01), CR_LA (ρ = 0.54, *p* ≤ 0.01), and Digit_BW (ρ = 0.40, *p* ≤ 0.01), as well as negative correlations between TMT_A (ρ = −0.38, *p* ≤ 0.05) and TMT_B (ρ = −0.33, *p* ≤ 0.05). Moreover, significant positive associations were found between flexibility and CR_S (ρ = 0.36, *p* ≤ 0.05), CR_WR (ρ = 0.50, *p* ≤ 0.01), CR_LA (ρ = 0.47, *p* ≤ 0.01), and Digit_FW (ρ = 0.42, *p* ≤ 0.01), while negative associations were found between TMT_A (ρ = −0.39, *p* ≤ 0.05) and TMT_B (ρ = −0.37, *p* ≤ 0.05). Also, significant positive correlations emerged between originality with CR_WR (ρ = 0.34, *p* ≤ 0.05), CR_LA (ρ = 0.51, *p* ≤ 0.01), and Digit_BW (ρ = 0.31, *p* ≤ 0.05), and between elaboration with CR_WR (ρ = 0.33, *p* ≤ 0.05), CR_LA (ρ = 0.45, *p* ≤ 0.01), and Digit_BW (ρ = 0.37, *p* ≤ 0.05).

#### 3.1.2. Milan

As reported in Table A2 (see Appendix A), a significant positive correlation was found between CR_WR and PWB_CS (ρ = 0.33, *p* ≤ 0.05). Regarding DT, a significant positive correlation emerged between fluency and CR_WR (ρ = 0.37, *p* ≤ 0.01) and a negative correlation with both TMT_B (ρ = −0.39, *p* ≤ 0.01) and TMT_B-A (ρ = −0.34, *p* ≤ 0.04). Moreover, significant positive correlations were found between flexibility with CR_S (ρ = 0.29, *p* ≤ 0.05) and CR_LA (ρ = 0.37, *p* ≤ 0.01), and between originality and CR_LA (ρ = 0.36, *p* ≤ 0.01). Also, a significant positive correlation emerged between elaboration with PWB_EC (ρ = 0.32, *p* ≤ 0.05), PWB_CS (ρ = 0.34, *p* ≤ 0.51), and CR_WR (ρ = 0.48, *p* ≤ 0.01). Finally, contrary to our expectations, significant positive correlations were found between PWB_PS with TMT_B (ρ = 0.30, *p* ≤ 0.05) and TMT_B-A (ρ = 0.41, *p* ≤ 0.01) and a negative correlation between elaboration and Digit_BW (ρ = 0.30, *p* ≤ 0.05).

#### 3.1.3. Cagliari

Significant negative correlations were found between CR_LA with TMT_B (ρ = 0.53, *p* ≤ 0.01) and TMT_B-A (ρ = 0.57, *p* ≤ 0.01), as shown in Table A3 (see Appendix A). Regarding DT, significant positive correlations were found between fluency and Digit_BW (ρ = 0.39, *p* ≤ 0.05), flexibility and Digit_BW (ρ = 0.39, *p* ≤ 0.05), and elaboration and Digit_FW (ρ = 0.37, *p* ≤ 0.05).

#### 3.1.4. Bergamo

As shown in Table A4 (see Appendix A), significant negative associations were reported between fluency with TMT_A (ρ = −0.45, *p* ≤ 0.05), TMT_B (ρ = −0.48, *p* ≤ 0.01), and TMT_B-A (ρ = −0.45, *p* ≤ 0.05), but positive associations were reported with CR_S (ρ = 0.47, *p* ≤ 0.01). Significant negative correlations were found between flexibility with TMT_A (ρ = −0.41, *p* ≤ 0.05), TMT_B (ρ = −0.48, *p* ≤ 0.01), and TMT_B-A (ρ = −0.50, *p* ≤ 0.01) and positive correlations were found with CR_S (ρ = 0.42, *p* ≤ 0.01). Also, significant negative correlations were found between originality with TMT_A (ρ = −0.40, *p* ≤ 0.05), TMT_B (ρ = −0.49, *p* ≤ 0.01), and TMT_B-A (ρ = −0.46, *p* ≤ 0.01) and positive correlations were found with CR_S (ρ = 0.40, *p* ≤ 0.05). Moreover, significant negative correlations were found between elaboration with TMT_B (ρ = −0.45, *p* ≤ 0.05) and TMT_B-A (ρ = −0.44, *p* ≤ 0.05) and positive correlations with CR_LA (ρ = 0.31, *p* ≤ 0.05) and Digit_BW (ρ = 0.33, *p* ≤ 0.05). Finally, contrary to our expectations, significant negative correlations were found between PWB_PS with TMT_B (ρ = −0.50, *p* ≤ 0.01), CR_WR (ρ = −0.33, *p* ≤ 0.05), fluency (ρ = −0.34, *p* ≤ 0.05), and flexibility (ρ = −0.49, *p* ≤ 0.01).

### 3.2. Generalized Linear Models

Several GLM models were used for each subscale to explore the relationship between the predictors and the grouping factor. The results for each predictor are reported below.

#### 3.2.1. Personal Well-Being

The analysis revealed a significant group effect for coping strategies (Wald Chi-square = 34, *p* < 0.001, η^2^ = 0.171), indicating differences in personal well-being scores across the groups (see Table 2).

Significant group effects for coping strategies (Wald Chi-square = 34, *p* ≤ 0.001, η^2^ = 0.171) were found, indicating differences in personal well-being scores across the groups. Post-hoc comparisons using the BZ as a reference group showed significant differences with BG (Mdif = 3.54, SE = 0.87, 95% CI = 1.25–5.84, pBonf ≤ 0.001) and MI (Mdif = 4.33, SE = 0.80, 95% CI = 2.21–6.45, pBonf ≤ 0.001). Moreover, comparisons using MI as a reference group reported significant differences with CA (Mdif = −2.82, SE = 0.94, 95% CI = −5.32–−0.32, pBonf = 0.018). A significant group effect was also observed for personal satisfaction (Wald Chi-square = 36.82, *p* ≤ 0.001, η^2^ = 0.182). Post-hoc tests revealed that the reference BZ group exhibited significantly higher scores compared to BG (Mdif = 5, SE = 1.06, 95% CI = 2.19–7.80; pBonf ≤ 0.001), MI (Mdif = 5.59, SE = 1.03, 95% CI = 2.87–8.30, pBonf ≤ 0.001), and CA (Mdif = 3.21, SE = 1.19, 95% CI = 0.67–6.36, pBonf = 0.048). Regarding emotion regulation skills, a significant group effect was found (Wald Chi-square = 21.12, *p* ≤ 0.001, η^2^ = 0.113). Post-hoc comparisons indicated that BZ scored higher compared to BG (Mdif = 3.62, SE = 1, 95% CI = 1.2–6.24, pBonf = 0.002) and MI (Mdif = 3.51, SE = 0.89, 95% CI = −0.98–4.40, pBonf ≤ 0.001).

#### 3.2.2. Cognitive Reserve

Results concerning CR are reported in Table 3.

The analysis revealed a significant group effect for leisure activities (Wald Chi-square = 11.31, *p* ≤ 0.01, η^2^ = 0.06), as well as a significant effect of age (Wald Chi-square = 11.67, *p* ≤ 0.001, η^2^ = 0.07). Post-hoc comparisons showed that MI participants reported significantly higher scores also compared to BG (Mdif = 15.10, SE = 4.98, 95% CI = 1.95–28.20, pBonf = 0.015). For work, a significant effect of age emerged (Wald Chi-square = 22.53, *p* ≤ 0.001, η^2^ = 0.12), indicating a strong association between age and scores on work. Moreover, a significant group effect was observed for school (Wald Chi-square = 36.34, *p* ≤ 0.001, η^2^ = 0.18), alongside a significant effect of age (Wald Chi-square = 9.10, *p* ≤ 0.01, η^2^ = 0.05). Post-hoc comparisons showed that BZ scored lower than BG (Mdif = −9.84, SE = 2.16, 95% CI = −15.54–−4.15, pBonf ≤ 0.001), MI (Mdif = −13.18, SE = 2.75, 95% CI = −20.43–−5.93, pBonf ≤ 0.001), and CA (Mdif = −10.70, SE = 3.14, 95% CI = −18.74–−2.43, pBonf = 0.004).

#### 3.2.3. Neuropsychological Battery

A significant group effect was observed for Digit Span Backward (Wald Chi-square = 8.30, *p* ≤ 0.05, η^2^ = 0.05) (see Table 4).

Post-hoc comparisons showed that CA scored significantly higher than MI (Mdif = 0.57, SE = 0.21, 95% CI = 0.03–1.13, pBonf = 0.048). For digit span forward, the group effect was also significant (Wald Chi-square = 10.12, *p* ≤ 0.05, η^2^ = 0.06): Participants from BZ had lower scores compared to BG (Mdif = −0.66, SE = 0.24, 95% CI = −1.30–−0.03, pBonf = 0.033). Moreover, BG scored significantly higher than CA (Mdif = 0.68, SE = 0.25, 95% CI = 0.02–1.33, *p* ≤ 0.001).

A significant group effect was detected for TMT_A (Wald Chi-square = 21.28, *p* ≤ 0.05, η^2^ = 0.11). Post-hoc comparisons revealed that participants from BZ scored higher than MI (Mdif = 22.57, SE = 5.10, 95% CI = −7.82–28.41, pBonf = 0.048). For TMT_B, a highly significant group effect was observed (Wald Chi-square = 23.90, *p* ≤ 0.001, η^2^ = 0.13). Post-hoc comparisons revealed that participants from BZ scored higher compared to BG (Mdif = 67.51, SE = 24.30, 95% CI = 3.42–131.61, pBonf = 0.033) and MI (Mdif = 80.81, SE = 20.07, 95% CI = 27.86–133.76, *p* ≤ 0.001). Also, TMT_B-A showed a significant group effect (Wald Chi-square = 17.48, *p* ≤ 0.001, η^2^ = 0.10). A post-hoc comparison revealed that BZ scored higher than MI (Mdif = 58.04, SE = 17.43, 95% CI = 12.05–104.04, pBonf ≤ 0.005).

#### 3.2.4. Depression, Anxiety, Stress Scale

Table 5 reports results considering DASS subscales as predictors.

The analysis revealed a significant group effect for depression (Wald Chi-square = 10.00, *p* ≤ 0.01, η^2^ = 0.06), indicating differences in scores across the groups. Post-hoc comparisons using BZ as a reference group showed that BZ scored lower than MI (Mdif = −1.49, SE = 0.52, 95% CI = −2.86–−0.11, pBonf = 0.026).

#### 3.2.5. Divergent Thinking

Table 6 reports results considering DT subscales as predictors.

The analysis revealed a group effect for fluency (Wald Chi-square = 7.53, *p* ≈ 0.05, η^2^ = 0.04). However, no post-hoc pairwise comparison reached a significant *p*-value. A significant group effect was found for flexibility (Wald Chi-square = 11.35, *p* ≤ 0.01, η^2^ = 0.06). Post-hoc analyses revealed that participants from BZ had higher scores than CA (Mdif = 2.31, SE = 0.81, 95% CI = 0.17–4.44, pBonf = 0.026). The group effect for originality was highly significant (Wald Chi-square = 21.65, *p* ≤ 0.001, η^2^ = 0.12), indicating substantial differences across groups. Post-hoc comparisons showed that participants from BZ had higher scores than CA (Mdif = 3.38, SE = 1.94, 95% CI = 3.28–13.50, pBonf ≤ 0.001). Also, CA reported significantly lower scores compared to BG (Mdif = −5.32, SE = 2, 95% CI = −10.61–−0.03, pBonf = 0.048) and MI (Mdif = −4.21, SE = 1.51, 95% CI = −8.20–−0.23, pBonf = 0.032). The group effect for elaboration was highly significant (Wald Chi-square = 48.90, *p* ≤ 0.001, η^2^ = 0.23), reflecting the largest effect size among the subscales. Post-hoc comparisons revealed that participants from BZ reported significantly lower scores compared to BG (Mdif = 3.62, SE = 0.99, 95% CI = 1–6.24, pBonf = 0.002) and MI (Mdif = 3.51, SE = 0.90, 95% CI = 1.16–5.87, pBonf ≤ 0.001).

## 4. Discussion

To identify factors that could promote successful aging and to explore potential interventions to slow cognitive decline, our study sought to investigate whether DT and CR—which have received limited attention in BZ (except [24,28] for CR)—differed from those observed in MI, BG, and CA. These areas are markedly distinct from BZ. The objective was to determine whether DT and CR contribute to successful aging in BZ and to explore the differences in these variables compared to other contexts. Additionally, the study aimed to examine the relationships between PWB, depressive indices, DT, CR, and cognitive functioning within each group. This approach aimed to identify group-specific characteristics and highlight the most significant relationships for each dimension within each group.

### 4.1. Exploring Determinants of Successful Aging in BZ

In our effort to explore the factors contributing to successful aging among the elderly in BZ, we sought to assess whether DT—investigated for the first time in this study within the BZ context—and CR, which has been minimally studied in BZ [24,28], demonstrated higher values compared to other population centers. We also examined the indices of cognitive functioning (including active and passive memory span, processing speed, visual scanning, number sequencing skills, shifting, and cognitive flexibility) and affective variables (PWB, emotional competence (EC), coping strategies (CS), personal satisfaction (PS), depressive indices, stress, and anxiety) to gain a more comprehensive understanding of the sample’s characteristics.

#### 4.1.1. Psychological Well-Being and Depressive Indices

In line with the literature, BZ had the highest levels of PWB (total index, EC, CS, and PS) and the lowest levels of depressive indices (depression, anxiety, and stress) [3,8,9,12,14,16,17,35,68,82], although the differences were not statistically significant for all variables. Notable differences were observed between BZ, MI, and BG for all PWB indices and depressive indices. However, no significant differences were found between BZ and CA for PWB, CS, EC, and depressive indices. Regarding stress, the only significant difference was between BZ and MI, while no significant differences were found for anxiety across the centers.

#### 4.1.2. Cognitive Reserve

As expected, the only significant difference in CR between BZ and the other centers was related to education, with BZ having the lowest CR for this component [83,84]. This suggests that BZ inhabitants derive greater benefits from work and leisure activities, which is consistent with the existing literature [24].

#### 4.1.3. Cognitive Functioning

Contrary to expectations based on the literature [8], BZ exhibited lower scores than other centers for passive memory span, visual-spatial search, and cognitive flexibility. A statistically significant difference emerged between BZ and MI for cognitive flexibility. Conversely, BZ showed comparable scores to other centers for active memory span.

However, cognitive functioning in BZ could be further investigated (see [24,85] for brief global cognitive function assessments; [28] for shifting and cognitive flexibility; [86] for comprehensive global evaluations). The lower CR for schooling may also contribute to these outcomes, and the instruments used in this study might not fully capture the cognitive functioning of BZ inhabitants.

#### 4.1.4. Divergent Thinking

Notably, when analyzing DT, a component of cognitive functioning [57,63,69,70,71], BZ stood out as the center with higher DT indices compared to other centers. Although the difference in DT fluency was not significant, significant differences emerged for DT originality and flexibility (compared to CA), as well as DT elaboration (compared to BG, which exhibited the lowest indices).

### 4.2. Implications

The findings suggest that, while BZ inhabitants have lower schooling-related CR, they benefit significantly from CR related to work and leisure activities, aligning with studies emphasizing the social and contextual characteristics of BZs [1,16,68,83,87]. Furthermore, the high DT indices observed in the BZ appear protective, influencing PWB and depressive indices. BZ was not only the area with the highest levels of PWB and the lowest levels of depression, anxiety, and stress but also the area with the highest levels of DT.

### 4.3. What Are the Relationships Between DT, CR, Cognitive Functioning, and PWB Within Each Different Context?

The results indicate that the relationships among DT, CR, cognitive functioning, and PWB vary across groups. Notably, the BZ group exhibited relationships among all variables, with more significant correlations than other centers. However, an exception was the lack of a direct relationship between DT and PWB in the BZ group, a relationship observed only in the MI and BG groups. In MI, slight correlations were found between DT (elaboration creativity index) and PWB components (CS and EC), while in BG the DT originality index showed a slight correlation with depressive components. These findings are particularly surprising for the BZ group due to its high DT and PWB indices.

#### 4.3.1. Relationships Between PWB and Other Variables

In the BZ group, PWB showed the most significant correlations with CR. Specifically, PWB EC correlated moderately with CR at work and leisure activities. PWB CS and depressive indices showed slight correlations with CR at work. In the MI group, slight correlations emerged between PWB CS and CR at work, but no other variables showed a correlation with PWB components or depressive indices. This suggests that the PS component of PWB, as well as indices of anxiety and stress, is unrelated to cognitive function, DT, and CR. These findings align with the questions of an indirect relationship between PWB and DT and a direct relationship between PWB and CR.

#### 4.3.2. DT, CR, and Cognitive Functioning Relationships

DT showed relationships with CR in the MI, BG, and BZ groups, as well as with cognitive functioning across all groups. Notably, in MI, CR correlated only with DT cognitive flexibility, while in BG the influence of CR was limited, isolated, and only weakly associated with DT. In contrast, in BZ, CR in both work and leisure contexts correlated with all DT indices, including fluency, flexibility, and originality, highlighting a stronger link between these variables compared to the other centers. This observation is particularly intriguing, as the BZ group is the only one to show moderate relationships between DT and CR in both leisure and work, along with a mild relationship in schooling. As previously described, CR was related to PWB and DT, while cognitive functioning was significantly associated with CR only in the BZ and CA groups: in BZ, with school- and work-related CR (for passive memory span (MBT) and cognitive flexibility); in CA, with leisure-related CR for cognitive flexibility. No relationships were found between CR and cognitive function in MI and BG. However, DT and cognitive function were correlated across all groups. These findings confirm that while CR supports cognitive functioning, it is also influenced by DT, which contributes to successful aging by enhancing various components of CR.

#### 4.3.3. Contextual Differences

The differences across centers reflect their unique characteristics. MI is a large urban center in Northern Italy that offers diverse opportunities for work, education, and professional growth; CA and BG are mid-sized centers, BG is in Lombardy (Northern Italy), like MI, while CA is in Sardinia and experiences limitations due to its insularity. In CA social connections are preserved in the neighborhoods where the support provided by third-sector organizations serves as an aid to the community [88]. BZ is composed of small, isolated centers in mountainous areas, where agro-pastoral work and collective social interactions dominate daily life. Community and neighborly support are pivotal in these settings [1,3,7,8,11,12,16]. These contextual differences likely explain why leisure-related CR correlates with DT in BZ but only weakly in BG and why leisure-related CR in CA correlates with cognitive flexibility. The centers, therefore, exhibit significant differences, resulting in specific abilities or indicators of ability holding greater importance than others in each center (for example, leisure CR compared to schooling in the BZs) for cognitive functioning, depressive indices, and psychological well-being.

### 4.4. Implications for Interventions

Our findings underline the importance of DT as a protective factor for cognitive functioning and successful aging across all centers. We believe these data are highly important for informing the choice of interventions to propose to promote active aging. While relationships between DT and PWB were observed in MI and BG, they were slight and isolated, failing to show the expected robustness. Conversely, the relationship between PWB and CR was more pronounced, suggesting that DT has an indirect positive effect on PWB through CR. DT supports coping strategies and emotional skills in older adults despite the natural aging process, which typically leads to a decline in the perception of one’s own emotional competencies [58,89,90]. These aspects should be carefully considered when designing interventions aimed at enhancing quality of life in later years [21]. Both DT and an active lifestyle contribute to the preservation of cognitive functions and the reduction of stress [91,92]. Strengthening DT could be particularly beneficial for older adults, especially those living in lonely or inactive environments, where social engagement and participation in life tend to be neglected [39]. Such interventions could improve quality of life by enhancing cognitive flexibility, emotional understanding, and social relationships, as well as fostering trust in others, motivation to share experiences, and confidence in expressing opinions in different contexts. Ultimately, this could lead to greater psychological well-being and lower levels of depressive symptoms. These findings emphasize the need for further investigation into DT’s role in successful aging, particularly in BZ.

This study has several limitations that should be considered when interpreting the results. First, it could be important to consider that the cognitive functioning evaluations might not completely cover the complexity of cognitive abilities in the elderly and that CRIq may not comprehensively assess all the domains of CR, and this could limit the complete generalisation of results. One potential limitation of this study is the voluntary enrollment of participants, which may have led to a self-selection bias. Participants who chose to take part in the study might differ from the general population in relevant ways, potentially affecting the generalizability of the findings. Future studies should consider adopting broader recruitment strategies to ensure a more representative sample. In methodological terms, the sampling process did not follow randomization criteria. Recruiting participants through an appropriate sampling method, with a prior calculation of the required sample size, would have ensured a greater robustness of the findings. To enhance the representativeness of future research, broader and more diverse sampling strategies should be considered. Recruiting participants from a variety of settings, including community centers, healthcare institutions, and non-voluntary recruitment pools, may help mitigate self-selection bias and improve the external validity of the findings. Finally, the sample size was insufficient to perform analyses with higher statistical power, such as parametric analyses, which require a normal data distribution and a sufficient number of participants to produce more reliable results. The use of non-parametric analyses, necessitated by the non-normal distribution of the data, may reduce statistical power compared to parametric approaches. However, the small sample size would have limited even a parametric analysis. Therefore, a larger and more balanced sample would have been crucial to enhance the statistical validity and generalizability of the findings. These considerations underscore the need for future studies to adopt more rigorous sampling methodologies and carefully plan the sample size to optimize the detection of significant effects and achieve more precise estimates. Finally, it might be useful to replicate this study by also investigating verbal DT, which, in previous studies, has already demonstrated overlaps with CR [53,54,60] and relationships with PWB [58] and cognitive functioning [57,63,69,70,71].

## Figures and Tables

**Table 1 brainsci-15-00249-t001:** Participants’ sociodemographic data according to the group.

	BZ	MI	CA	BG	*p*-Value
N	41	50	32	42	0.273 ^a^
Age mean (SD; range)	74.41 (6.80; 28)	73.98 (6.68; 23)	72.75 (6.66; 25)	73.74 (6.70; 23)	0.720
Gender					0.061
*male*	12 (7.3%)	21 (12.7%)	19 (11.5%)	21 (12.7%)	
*female*	29 (17.6%)	29 (17.6%)	13 (7.9%)	21 (12.7%)	
Average years of education (SD; range)	8.61 (2.9; 11)	11.3 (4.81; 18)	12.1 (6.9; 24)	10.33 (3.2; 13)	0.024

Note: ^a^ = exact test of significance; BZ = Blue Zone; MI = Milan city; CA = Cagliari city; BG = Bergamo city.

**Table 2 brainsci-15-00249-t002:** Wald Chi-square statistics and effect sizes (η^2^) for PWB subscales are reported, along with model’s estimated means.

GLM Model Results		Tests of Within-Subject Effects			Model’s Estimated Means		
Subscales	Effects	Wald Chi-Square	df	η^2^	Group	Mean	SE (95% CI)
PWB_CS	Group	34 ***	3	0.171	BZ	29.95	0.68 (25.08; 27.73)
					MI	25.62	0.59 (24.47; 26.77)
					CA	28.44	0.74 (26.98; 29.90)
					BG	26.41	0.55 (28.87; 31.03)
PWB_PS	Group	36.82 ***	3	0.182	BZ	38.80	0.67 (37.49; 40.12)
					MI	33.22	0.78 (31.69; 34.75)
					CA	35.59	0.99 (33.66; 37.53)
					BG	33.81	0.83 (32.19; 35.43)
PWB_EC	Group	21.12 ***	3	0.113	BZ	34.17	0.57 (33.05; 35.29)
					MI	30.66	0.69 (29.32; 32.00)
					CA	32.47	0.84 (30.83; 34.11)
					BG	30.55	0.81 (28.96; 32.13)

Note: *** = *p* ≤ 0.001; PWB_CS = coping strategies; PWB_PS = personal satisfaction; PWB_EC= emotion regulation skills; BZ = Blue Zone; MI = Milan city; CA = Cagliari city; BG = Bergamo city; η^2^ = eta-squared.

**Table 3 brainsci-15-00249-t003:** Wald Chi-square statistics and effect sizes (η^2^) for cognitive reserve subscales are reported, along with model’s estimated means.

GLM Models Results		Tests of Within-Subject Effects			Model’s Estimated Means		
Subscales	Effects	Wald Chi-Square	df	η^2^	Group	Mean	SE (95% CI)
CR_LA	Group	11.31 **	3	0.06	BZ	121.42	3.11 (115.33; 127.50)
	Age	11.67 ***	1	0.07	MI	127.77	2.77 (122.34; 133.19)
					CA	115.62	4.35 (107.09; 124.14)
					BG	112.69	4.03 (104.80; 120.58)
					age		
CR_WR	Group	2.39	3	0.01	BZ	103.74	2.56 (98.72; 108.77)
	Age	22.53 ***	1	0.12	MI	103.08	2.83 (97.53; 108.64)
					CA	99.96	3.55 (93.00; 106.93)
					BG	106.78	2.75 (101.39; 112.17)
					age		
CR_S	Group	36.34 ***	3	0.18	BZ	97.32	1.36 (94.66; 99.99)
	Age	9.10 **	1	0.05	MI	110.50	2.37 (105.86; 115.15)

Note: *** = *p* ≤ 0.001; ** = *p*≤ 0.01; CR_LA = CRIq leisure activities; CR_WR = CRIq work; CR_S = CRIq school; BZ = Blue Zone; MI = Milan city; CA = Cagliari city; BG=Bergamo city; η^2^ = eta-squared.

**Table 4 brainsci-15-00249-t004:** Wald Chi-square statistics and effect sizes (η^2^) for neuropsychological sub-tests are reported, along with model’s estimated means.

GLM Model Results		Tests of Within-Subject Effects			Model’s Estimated Means		
Subscales	Effects	Wald Chi-Square	df	η^2^	Group	Mean	SE (95% CI)
Digit_BW	Group	8.30 *	3	0.05	BZ	4.42	0.14 (4.14; 4.69)
					MI	4.39	0.13 (4.13; 4.65)
					CA	4.96	0.17 (4.63; 5.29)
					BG	4.56	0.17 (4.23; 4.88)
Digit_FW	Group	10.12 *	3	0.06	BZ	5.65	0.16 (5.34; 5.96)
					MI	5.95	0.14 (5.68; 6.22)
					CA	5.64	0.17 (5.30; 5.98)
					BG	6.32	0.18 (5.96; 6.67)
TMT_A	Group	21.28 *	3	0.11	BZ	51.24	4.53 (42.36; 60.13)
					MI	28.72	2.35 (24.12; 33.32)
					CA	40.95	5.16 (30.84; 51.05)
					BG	34.66	4.93 (24.99; 44.32)
TMT_B	Group	23.90 ***	3	0.13	BZ	158.06	19.35 (120.13; 195.98)
					MI	77.25	5.34 (66.78; 87.72)
					CA	126.75	14.98 (97.38; 156.11)
					BG	90.54	14.69 (61.75; 119.33)
TMT_B-A	Group	17.48 ***	3	0.10	BZ	112.15	16.84 (79.14; 145.15)
					MI	54.10	4.52 (45.25; 62.96)
					CA	91.90	12.71 (67.00; 116.80)
					BG	59.78	10.89 (38.45; 81.12)

Note: *** = *p* ≤ 0.001; * = *p* ≤ 0.05; Digit_BW = digit span backward; Digit_FW = digit span forward; TMT_A = trail making test: visual searching; TMT_B = trail making test: divided attention; TMT_B-A = trail making test: attentional shifting; BZ = Blue Zone; MI = Milan city; CA = Cagliari city; BG = Bergamo city; η^2^ = eta-squared.

**Table 5 brainsci-15-00249-t005:** Wald Chi-square statistics and effect sizes (η^2^) for DASS subscales are reported, along with model’s estimated means (coefficients are in log-odds).

GLM Model Results		Tests of Within-Subject Effects			Model’s Estimated Means		
Subscales	Effects	Wald Chi-Square	df	η^2^	Group	Mean	SE (95% CI)
DASS_dep	Group	10.00 **	3	0.06	BZ	2.63	0.29 (2.13; 3.26)
					MI	4.12	0.43 (3.35; 5.07)
					CA	3.63	0.66 (2.54; 5.17)
					BG	3.83	0.40 (3.12; 4.71)
DASS_anxiety	Group	1.60	3	0.01	BZ	3.34	0.41 (2.63; 4.25)
					MI	3.46	0.34 (2.86; 4.19)
					CA	3.16	0.44 (2.40; 4.15)
					BG	3.90	0.44 (3.13; 4.87)
DASS_stress	Group	4.84	3	0.03	BZ	4.34	0.49 (3.48; 5.41)
					MI	5.84	0.50 (4.94; 6.90)
					CA	4.81	0.71 (3.60; 6.44)
					BG	5.38	0.48 (4.52; 6.40)

Note: ** = *p* ≤ 0.01; DASS_dep = depression; DASS_anxiety = anxiety; DASS_stress = stress; BZ = Blue Zone; MI = Milan city; CA = Cagliari city; BG = Bergamo city; η^2^ = eta-squared.

**Table 6 brainsci-15-00249-t006:** Wald Chi-square statistics and effect sizes (η^2^) for DT subscales are reported, along with model’s estimated means.

GLM Model Results		Tests of Within-Subject Effects			Model’s Estimated Means		
Subscales	Effects	Wald Chi-Square	df	η^2^	Group	Mean	SE (95% CI)
Fluency	Group	7.53 +	3	0.04	BZ	12.05	0.91 (10.27; 13.83)
					MI	11.30	0.71 (9.90; 12.70)
					CA	9.16	0.74 (7.71; 10.61)
					BG	11.29	1.06 (9.21; 13.36)
Flexibility	Group	11.35 **	3	0.06	BZ	9.02	0.61 (7.83; 10.22)
					MI	8.80	0.51 (7.79; 9.81)
					CA	6.72	0.53 (5.68; 7.76)
					BG	8.60	0.70 (7.23; 9.96)
Originality	Group	21.65 ***	3	0.12	BZ	18.63	1.62 (15.45; 21.82)
					MI	14.46	1.08 (12.34; 16.58)
					CA	10.25	1.05 (8.18; 12.32)
					BG	15.57	1.71 (12.23; 18.92)
Elaboration	Group	48.90 ***	3	0.23	BZ	15.29	1.46 (12.43; 18.15)
					MI	16.56	1.47 (13.68; 19.44)
					CA	11.63	1.28 (9.11; 14.14)
					BG	6.52	0.86 (4.84; 8.21)

Note: *** = *p* ≤ 0.001; ** = *p* ≤ 0.01; + = *p* = 0.057; BZ = Blue Zone; MI = Milan city; CA = Cagliari city; BG = Bergamo city; η^2^ = eta-squared.

## Data Availability

The dataset will be made available upon request due to privacy and ethical restrictions.

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
