# Peer review of "The Role of Well-Being, Divergent Thinking, and Cognitive Reserve in Different Socio-Cultural Contexts"

_brainsci, 2025, doi:10.3390/brainsci15030249_

Round 1
Reviewer 1 Report
Comments and Suggestions for Authors
This manuscript investigate the relationships between DT/CR/PWB/memory functions/depression/stress/anxiety indexes across BZ and several major Italian cities. However, several issues need to be addressed:
1. The voluntary enrollment usually results in self-selection bias, that is, those who are more willing to join the study might have differences compared to the general population.
2. Although the authors applied well-known instruments (e.g. the MMSE, CRIq, etc..), it is important to take into consideration that the cognitive functioning evaluation might not completely cover the complexity of cognitive abilities in the elderly. For example, the MMSE has its limitations in detecting subtle cognitive changes and the CRIq may not comprehensively assess all the domains of CR.
3. The sample size is relatively small across four groups.
4. Correlational analyses are overinterpreted as causal pathways, and the absence of mediation/moderation analyses weakens theoretical claims.
5. Remove labels like “near-significant” to avoid misleading interpretations.
Author Response
- The voluntary enrollment usually results in self-selection bias, that is, those who are more willing to join the study might have differences compared to the general population.
We understand the concern regarding potential self-selection bias due to the voluntary enrollment of participants. his limitation was already discussed in the limitations section of the manuscript (page 16, paragraph 4.4, lines 634-646). To improve clarity, we have expanded this section by further specifying the possible implications of self-selection bias and suggesting future research strategies to adopt broader and more representative sampling methods.
- Although the authors applied well-known instruments (e.g. the MMSE, CRIq, etc..), it is important to take into consideration that the cognitive functioning evaluation might not completely cover the complexity of cognitive abilities in the elderly. For example, the MMSE has its limitations in detecting subtle cognitive changes and the CRIq may not comprehensively assess all the domains of CR.
We have taken your suggestion into account by including this section in the limit of study, (page 16, paragraph 4.4, lines 631-634).
- The sample size is relatively small across four groups.
We acknowledge that the sample size is relatively small and this was already reported in the methodological section and the limitations section of the manuscript. However, the statistical analyses applied (GLMs with corrections for assumption violations) allowed us to obtain robust results. To further clarify this aspect, we have emphasized in the revised manuscript the need for future studies with larger and more representative samples (page 16, paragraph 4.4, lines 642-646) to enhance the generalizability of the findings;in methodological section the need to use GLM models (page 6, paragraph 2.4, lines 297-300) was mentioned as well.
- Correlational analyses are overinterpreted as causal pathways, and the absence of mediation/moderation analyses weakens theoretical claims.
Thank you for your suggestion. We are aware that correlations do not imply causation. Throughout the manuscript, we have consistently indicated that our analyses identify associations between variables rather than causal relationships. To prevent any potential misunderstanding, we have further clarified this distinction in the results section (page 7, paragraph 3.1, lines 323–327), explicitly stating that the observed relationships suggest patterns that would require longitudinal studies or mediation/moderation models to investigate possible causal directions.
Remove labels like “near-significant” to avoid misleading interpretations.
Thank you for your suggestion. We acknowledge the importance of avoiding potentially misleading interpretations regarding statistical significance. In the revised manuscript, we have removed all instances of the term “near-significant” and similar expressions. Instead, we have provided a more precise interpretation of the statistical results by directly reporting the p-values and confidence intervals where applicable (page 13, paragraph 3.2.5).
Reviewer 2 Report
Comments and Suggestions for Authors
Thank you for the opportunity to review the manuscript.
The topic of the research is important and may be of interest. I have comments that may improve the manuscript.
1) The title of the article includes the word "creativity", however, the focus in the article is on DT. The concept of creativity in the title should be changed and refer to DT (even if it is feature of creativity).
Introduction:
2) While CR is well-defined with references to established research, DT is introduced without a clear theoretical framework.
3) The introduction claims that “no direct evidence yet links creativity to PWB,” yet later, it suggests that DT mediates the relationship between CR and PWB.
If DT indirectly affects PWB, this needs clarification—does DT independently influence PWB, or does it only do so via CR?
4) The Authors included Hypotheses. However, it seems to be study questions.
5) Reliability and validity data for all tools should be provided.
6) The results section is hard to follow, in particular, the first section.
Author Response
The title of the article includes the word "creativity", however, the focus in the article is on DT. The concept of creativity in the title should be changed and refer to DT (even if it is feature of creativity).
Thank you for your advice. We have made the change in the title as suggested.
While CR is well-defined with references to established research, DT is introduced without a clear theoretical framework.
Thank you for your suggestion. We have integrated the theoretical framework of DT, starting from creative thinking, its relationship with DT, and the effects these skills have on the ageing process. You can find the additions on page 2,3 paragraph 1, and line 83-108.
The introduction claims that “no direct evidence yet links creativity to PWB,” yet later, it suggests that DT mediates the relationship between CR and PWB. If DT indirectly affects PWB, this needs clarification—does DT independently influence PWB, or does it only do so via CR?
We have clarified the concept by explaining that DT and PWB have an indirect relationship through CR. You can find the additions on page 3, paragraph 1 and 1.1., and lines 123-124 and 142-143.
The Authors included Hypotheses. However, it seems to be study questions.
We have accepted your suggestion by changing the hypotheses to study questions. You can see the modifications on page 4, paragraph 1.2., and line 162.
Reliability and validity data for all tools should be provided.
We have included reliability and validity data for all the instruments used. You can see the modifications on pages 5-6, paragraph 2.3, lines 218-220, 230-231, 241, 252, and 261-263.
The results section is hard to follow, in particular, the first section.
We acknowledge the importance of presenting the results in a clear and structured manner. The first section of the Results was already organized to provide a logical progression of findings; however, to enhance readability, we have revised this section to improve clarity. In particular, we have slightly reorganized the structure by introducing clearer subheadings to better distinguish different types of analyses. Additionally, we have streamlined the explanations to ensure that statistical findings are presented in a more accessible way. To further improve readability, we have also refined the wording of key results, making sure that all reported effects are explicitly connected to the research questions. These modifications aim to facilitate the comprehension of the results while maintaining the rigor and integrity of the statistical reporting (page 7, paragraphs 3.1.1, 3.1.2, 3.1.3, 3.1.4, lines 333–386).
Reviewer 3 Report
Comments and Suggestions for Authors
Dear Authors,
I read your manuscript and it was an interesting reading, however I think you could strengthen a few aspects of the paper:
1. Since the sampling wasn't random, I'd like to see some discussion of potential biases and how they might affect your conclusions.
2. I think you should provide more information about the statistical choices you made, specifically why you opted for non-parametric analyses and what that means for interpreting the results.
3. You've identified interesting relationships between divergent thinking and well being, how might this inform programs to support healthy aging in different contexts?
4. i felt the discussion a bit overwhelming at times. Consider focusing more deeply on the key relationships that matter most for promoting well being in aging populations.
Author Response
Since the sampling wasn't random, I'd like to see some discussion of potential biases and how they might affect your conclusions.
We have already discussed the potential impact of non-random sampling in the manuscript. To make this information more explicit, we have expanded the limitations section (page 16, paragraph 4.4, lines 634–639, 642-646), adding further details on how this sampling method may have influenced the results. Additionally, we have suggested possible future approaches to mitigate this bias, such as employing more diverse and representative sampling strategies.
I think you should provide more information about the statistical choices you made, specifically why you opted for non-parametric analyses and what that means for interpreting the results.
Thank you for your suggestion. We acknowledge the importance of clearly justifying the choice of statistical analyses. he rationale for using Generalized Linear Models (GLMs) and non-parametric approaches was already provided in the methods section of the manuscript (page 6, paragraph 2.4, lines 297–315). Specifically, we stated that normality was assessed using Q-Q plots and additional assumptions such as homoscedasticity and collinearity were verified to ensure the robustness of the models. Furthermore, we explained that GLMs with normal distributions and identity link functions were applied where assumptions were met, while for variables violating normality, GLMs with gamma distributions and either identity or log link functions were used based on the results from the distribution fitting process. To further strengthen our models, Maximum Likelihood Estimation (MLE) with robust covariance matrices was employed to manage heteroscedasticity. Additionally, pairwise comparisons of estimated marginal means were performed with Bonferroni adjustments to control for multiple comparisons.To improve readability, we have slightly reorganized this section to ensure that the justification of the statistical approach is as explicit as possible.
You've identified interesting relationships between divergent thinking and well being, how might this inform programs to support healthy aging in different contexts?
Thank you for your suggestion. Our study are might be relevant for identifying effective interventions to promote successful ageing. Therefore, we have incorporated your suggestion by specifying how such interventions can be beneficial in different contexts and situations. You can find the integration on page 16, paragraph 4.4, lines from 614 to 629.
I felt the discussion a bit overwhelming at times. Consider focusing more deeply on the key relationships that matter most for promoting well being in aging populations.
We understand how important it is to have a text that is easy to read and we acknowledge that discussing so many variables can feel overwhelming. However, this was a deliberate choice to ensure a comprehensive analysis of all the aspects investigated. Nonetheless, we have taken your advice into account and have tried to streamline the discussion regarding variables related to contexts other than BZ (page 15, paragraph 4.3.2, lines 577-592)
Round 2
Reviewer 1 Report
Comments and Suggestions for Authors
The author has effectively addressed the questions posed. Publication is recommended.